# Initial Step of Selenite Reduction via Thioredoxin for Bacterial Selenoprotein Biosynthesis

**DOI:** 10.3390/ijms222010965

**Published:** 2021-10-11

**Authors:** Atsuki Shimizu, Ryuta Tobe, Riku Aono, Masao Inoue, Satoru Hagita, Kaito Kiriyama, Yosuke Toyotake, Takuya Ogawa, Tatsuo Kurihara, Kei Goto, N. Tejo Prakash, Hisaaki Mihara

**Affiliations:** 1College of Life Sciences, Ritsumeikan University, 1-1-1 Nojihigashi, Kusatsu 525-8577, Shiga, Japan; 0229atk33@gmail.com (A.S.); tober0925@gmail.com (R.T.); raono@fc.ritsumei.ac.jp (R.A.); mainoue@fc.ritsumei.ac.jp (M.I.); satoru.hagita@daikin.co.jp (S.H.); kirikaiw@gmail.com (K.K.); 2R-GIRO, Ritsumeikan University, 1-1-1 Nojihigashi, Kusatsu 525-8577, Shiga, Japan; 3Institute for Chemical Research, Kyoto University, Gokasho, Uji 611-0011, Kyoto, Japan; toyotake@fc.ritsumei.ac.jp (Y.T.); ogawa.tky@mbc.kuicr.kyoto-u.ac.jp (T.O.); kurihara@scl.kyoto-u.ac.jp (T.K.); 4Department of Chemistry, School of Science, Tokyo Institute of Technology, 2-12-1 Ookayama, Meguro-ku, Tokyo 152-8551, Japan; goto@chem.titech.ac.jp; 5School of Energy and Environment, Thapar Institute of Engineering and Technology, Patiala 147004, Punjab, India; ntejoprakash@thapar.edu

**Keywords:** bacteria, selenite, selenium delivery system, selenoprotein, thioredoxin

## Abstract

Many organisms reductively assimilate selenite to synthesize selenoprotein. Although the thioredoxin system, consisting of thioredoxin 1 (TrxA) and thioredoxin reductase with NADPH, can reduce selenite and is considered to facilitate selenite assimilation, the detailed mechanism remains obscure. Here, we show that selenite was reduced by the thioredoxin system from *Pseudomonas stutzeri* only in the presence of the TrxA (PsTrxA), and this system was specific to selenite among the oxyanions examined. Mutational analysis revealed that Cys33 and Cys36 residues in PsTrxA are important for selenite reduction. Free thiol-labeling assays suggested that Cys33 is more reactive than Cys36. Mass spectrometry analysis suggested that PsTrxA reduces selenite via PsTrxA-SeO intermediate formation. Furthermore, an in vivo formate dehydrogenase activity assay in *Escherichia coli* with a gene disruption suggested that TrxA is important for selenoprotein biosynthesis. The introduction of PsTrxA complemented the effects of TrxA disruption in *E. coli* cells, only when PsTrxA contained Cys33 and Cys36. Based on these results, we proposed the early steps of the link between selenite and selenoprotein biosynthesis via the formation of TrxA–selenium complexes.

## 1. Introduction

Selenium is an essential trace element in many organisms [1,2,3]. Most of its important roles in cells are exerted as the 21st amino acid selenocysteine (Sec) [4], which is translationally incorporated into selenoproteins such as formate dehydrogenase (FDH), glycine reductase, and hydrogenase in bacteria as well as glutathione peroxidases, selenoprotein P, and thioredoxin reductase (TXNRD) in mammals [5,6]. Bacterial selenoproteins mostly function in anaerobic energy metabolism, while those of mammals generally play antioxidant roles [5,6]. Compared with cysteine, that contains a thiol group, Sec with a selenol group is more nucleophilic and, thus, often serves as a catalytic residue in enzymes with redox activity [7].

In bacteria, selenide with ATP and water is converted to selenophosphate together with AMP and phosphate by selenophosphate synthetase (SPS) for Sec synthesis [8]. Seryl-tRNA^Sec^, formed by the aminoacylation of tRNA^Sec^ with serine by seryl-tRNA synthetase, is nucleophilically attacked by selenophosphate through the catalysis of selenocysteine synthase (SelA), resulting in selenocysteyl-tRNA^Sec^ generation [9]. Sec is incorporated into selenoproteins at UGA codons [10,11,12,13]. The specific translation elongation factor, SelB, delivers selenocysteyl-tRNA^Sec^ to the ribosome by recognizing the Sec insertion sequence (SECIS) located immediately downstream of the UGA codon on the mRNA. Mammalian selenoprotein synthetic machinery is slightly different from that of bacteria. First, seryl-tRNA^Sec^ is further converted to *O*-phosphoseryl-tRNA^Sec^ by *O*-phosphoseryl-tRNA ^Sec^ kinase, then selenocysteyl-tRNA^Sec^ is produced by Sep-tRNA:Sec-tRNA synthase using selenophosphate [14]. Second, mammalian SECIS is present in the 3’-untranslated region, and the SelB recognizes SECIS via a SECIS binding protein [6].

Although selenium plays essential roles in many organisms, it is toxic when present in excess [1,2,15]. Therefore, selenium delivery systems have been proposed to sequester toxic selenium intermediates to utilize the toxic element [16]. Glutathione (GSH) system and/or thioredoxin system (Trx system) are supposed to be involved in the reduction of selenite to selenide via NADPH [17,18,19]. GSH is the most abundant thiol in many organisms including *Escherichia coli* [20], and it is considered a major facilitator of selenite assimilation. However, selenium and GSH metabolites such as glutathione selenotrisulfide, are unstable [21]. Moreover, GSH is not found in most Gram-positive bacteria [20], and GSH reductase knockout mutants of *E. coli* produce a selenoprotein, FDH, suggesting that selenite assimilation via the GSH system is not a universal mechanism [22]. In contrast to GSH reductase, gene disruption of thioredoxin reductase (TrxR) decreases the FDH activity of *E. coli* cells [22]. TrxR, with thioredoxin 1 (TrxA) and NADPH, comprises the Trx system, which appears to be ubiquitous in many bacteria and participates in various redox reactions [23,24]. TrxA reduces oxidized substrate using two vicinal Cys residues in the active center, and the oxidized form of TrxA is reduced by TrxR using NADPH [23]. TXNRD can directly reduce selenite to selenide without mammalian thioredoxin, whereas that from *E. coli* cannot act without TrxA [25].

Although the previous study using ^75^Se-labeled selenite demonstrated that TrxA was labeled with ^75^Se even in the absence of TrxR [26], the mechanism of selenite reduction by TrxA has not been established. In this study, we focused on the reaction between selenite and TrxA from *Pseudomonas stutzeri* F2a [27] (PsTrxA), which is 70% identical to that of *E. coli* (EcTrxA). Since *P. stutzeri* F2a was isolated from seleniferous soil, the strain may serve as an interesting model for studying bacterial selenium metabolism. We observed the formation of a PsTrxA–SeO complex, implicating the early steps of selenite reduction via TrxA as a selenium delivery system in bacteria.

## 2. Results

### 2.1. Reduction Activity of the Trx System from P. stutzeri

The insulin disulfide-reductive cleaving activity of PsTrxA with dithiothreitol (DTT) was evaluated using the method previously described [28]. The turbidity of the reaction mixture increased due to the fact of precipitation of the free insulin B chain (Figure 1A), showing that PsTrxA reduced the disulfide bonds in insulin. The Cys33 and Cys36 residues of PsTrxA are broadly conserved in other TrxAs (Appendix A). To examine the involvement of these Cys residues in the disulfide-reducing activity, we constructed the PsTrxA mutants, C33A, C36A, and C33A/C36A, in which each Cys was substituted by Ala, and measured their activities. Unlike the wild-type PsTrxA, the mutants did not reduced insulin (Figure 1A), suggesting that the Cys residues are important active site residues in PsTrxA.

We next examined the selenite reduction activity of the Trx system using PsTrxA and TrxR from *P. stutzeri* F2a (PsTrxR) (Figure 1B). The Trx system with the wild-type PsTrxA exhibited selenite reduction activity, whereas that with the PsTrxA mutants did not, indicating that the active site Cys residues of PsTrxA play an essential role in the selenite reduction activity. These results are consistent with the previous findings that selenite is not reduced by bacterial TrxR alone, and that TrxA is required for selenite reduction [25]. We also examined whether the Trx system reduced other oxyanions, such as selenate, sulfite, sulfate, thiosulfate, nitrite, and nitrate. However, none of them served as a substrate (Appendix A). These results indicated that the Trx system is specific to selenite among the oxyanions examined in this study.

### 2.2. Number of Free Thiols in PsTrxA Incubated with Selenite

In the Trx system, selenite may be reduced by the reduced form of TrxA, resulting in the formation of selenide and oxidized TrxA with an intramolecular disulfide bond, which is re-converted to the reduced form by TrxR. Tamura et al. reported that EcTrxA was radiolabeled with ^75^Se derived from [^75^Se] selenite [26], implying the formation of a selenium-bound TrxA intermediate. In the GSH system, GSSeSG is formed by binding of selenium to the thiol groups of GSH [17]. We speculated that a somewhat similar selenium-bound intermediate could also occur in the Trx system, and that selenium would bind to TrxA via the thiol groups of the Cys residues. We further examined the involvement of the thiol groups of PsTrxA in selenite reduction by gel retardation assays using maleimide-conjugated polyethylene glycol (PEG-PCMal), which labels cysteine-thiol groups. Reduced PsTrxA was incubated with a five-fold molar excess of selenite, labeled with PEG-PCMal, and resolved by sodium dodecyl sulfate (SDS)-polyacrylamide gel electrophoresis (PAGE) (Figure 2).

The molecular mass of the wild-type and mutant PsTrxA proteins without PEG-PCMal labeling did not significantly differ according to SDS-PAGE (Figure 2). In contrast, various bands shifted in SDS-PAGE when proteins were labeled with PEG-PCMal. Based on the fact that the wild-type PsTrxA has three Cys residues, Cys33, Cys36, and Cys57 (Appendix A), whereas C33A and C36A has two and C33A/C36A has one, the changes in molecular mass apparently corresponded to the number of thiol groups labeled by PEG-PCMal.

In contrast, when the wild-type PsTrxA incubated with selenite was labeled with PEG-PCMal, the band shift diminished, and the molecular mass corresponded to the PsTrxA protein labeled with only one PEG-PCMal, indicating that the two thiol groups were not labeled (Figure 2). This result could be explained by the loss of reactivity between PEG-PCMal and the Cys residues due to the fact of their oxidation or modification with selenite. A weak band corresponding to the protein labeled with two PEG-PCMal was observed. This may be a non-specific product, because the identical band was also seen in the Cys33A/Cys36A, which has only one Cys residue, after incubation with PEG-PCMal in the absence of selenite.

We also incubated PsTrxA mutants with selenite, then performed a PEG-PCMal labeling assay (Figure 2). The molecular masses of most C33A and C33A/C36A were not changed, irrespective of selenite. In contrast, a large portion of C36A incubated with selenite was labeled with only one PEG-PCMal, whereas the mutant without selenite was labeled with two PEG-PCMal. Since Cys32 of EcTrxA corresponding to Cys33 of PsTrxA has a lower p*K*_a_ and is more reactive [29], the thiol group of Cys33 might attack selenite nucleophilically to produce a thioselenite moiety (–S-SeO_2_^−^) as previously suggested [26], then the thiol group would not be labeled with PEG-PCMal. The lower bands were also observed in C33A and C33A/C36A incubated with selenite followed by PEG-PCMal labeling. These bands were most likely due to the non-specific binding of reactive selenite to Cys residues, then the thiol group(s) would not be labeled with PEG-PCMal.

### 2.3. Formation of PsTrxA Complex with Selenium

We assumed that selenite oxidized or modified PsTrxA to prevent labeling with PEG-PCMal. To gain insight into the molecular state of PsTrxA reacted with selenite, the molecular mass of PsTrxA was analyzed using electrospray ionization (ESI)-mass spectrometry (MS) (Figure 3). A predominant protein species with a molecular mass of 13,839 Da corresponded with the recombinant PsTrxA lacking the N-terminal Met, which was calculated to be 13,837 Da from its amino acid sequence (Figure 3A). The intact PsTrxA with the N-terminal Met was also observed as a second major species with 13,970 Da, which is close to the calculated mass of 13,968 Da. When DTT-reduced PsTrxA was incubated with a five-fold molar excess of selenite, a protein species with a mass of 13,936 Da appeared as a new second major peak, while PsTrxA without the N-terminal Met (13,839 Da) remained predominant (Figure 3B). The shift of 97 Da was larger than the mass of selenium (79 Da), but close to that of SeO (95 Da). These data suggested that selenium bound to a significant fraction of PsTrxA in the form of SeO upon reaction with selenite.

### 2.4. Involvement of PsTrxA in Selenoprotein Synthesis

To explore whether TrxA is involved in delivering selenide for selenoprotein biosynthesis in vivo, the activity of the selenoprotein FDH was assayed in whole *E. coli* cells anaerobically cultured on solid medium using benzyl viologen as previously described [22]. The benzyl viologen assay directly reflects FDH activity in cells and indirectly reflects selenoprotein biosynthetic activity. Figure 4 shows that the wild-type *E. coli* cells stained purple, indicating FDH activity, whereas the cells with a disrupted SelA gene (Δ*selA*) were not stained as they could not express selenoproteins. *E. coli* with a disrupted EcTrxA gene (Δ*EctrxA*) had low levels of activity, suggesting that EcTrxA is a major facilitator of selenoprotein biosynthesis in this bacterium. Introducing the wild-type PsTrxA gene into the Δ*EctrxA* strain recovered FDH activity (Figure 4), suggesting that PsTrxA can complement the deletion of EcTrxA. In contrast, introducing the PsTrxA mutants, C33A, C36A, and C33A/C36A, did not complement EcTrxA disruption. These results suggest that Cys33 and Cys36 residues in PsTrxA are important for selenoprotein biosynthesis.

## 3. Discussion

Selenite can serve as a nutritional source of selenium for bacteria. Selenite is also provided from another inorganic selenium source, selenate, by selenate reductases such as *E. coli* YnfEFGH [30]. Selenite is then reduced to selenide in cells. Selenophosphate, which is essential for selenoprotein biosynthesis, is produced from selenide, ATP, and water by SPS [8]. However, since the *K*_m_ values of SPS for selenide reside in the toxic range (20–46 μM) for many organisms [31,32], it has been thought that selenium delivery systems may sequester the toxic element [16]. Some proteins, such as rhodanese, glyceraldehyde-3-phosphate dehydrogenase, and 3-mercaptopyruvate sulfurtransferase, are able to bind selenium and, therefore, debated as possible candidates for selenium delivery proteins [33,34]. However, their physiological relevance to selenium assimilation remains unclear.

The Trx system, which is distributed in many bacterial phyla, appeared as a promising candidate for selenium delivery system in bacteria [16]. The Cys33 and Cys36 residues in PsTrxA are broadly conserved in TrxAs (Appendix A). Disrupting these residues resulted in the loss of not only insulin, but also selenite reduction activity (Figure 1), suggesting that the conserved Cys residues are important for selenite reduction. The results of gel-shift assays suggested that Cys33 and Cys36 were oxidized and/or modified with selenite (Figure 2). A comparison of the two Cys residues suggested that Cys36 was not reactive to selenite without Cys33, due to the fact having less reactivity than Cys33. The ESI-MS analysis suggested that a specific fraction of PsTrxAs proteins formed a PsTrxA–SeO complex, which might be an intermediate in the initial step of selenite reduction by PsTrxA (Figure 3). The results of the benzyl viologen assays indicated that disrupted EcTrxA in *E. coli* led to a decrease in FDH activity (Figure 4). Considering the previous report that the disruption of TrxR in *E. coli* results in a decrease in FDH activity [22], the Trx system functions as the main selenium delivery system from selenite to selenoprotein synthesis in *E. coli*. In addition, since the Cys33 and Cys36 mutants did not complement EcTrxA gene deletion, these residues are important for selenoprotein biosynthesis from selenite in vivo. Taken together, these results indicate that the Trx system actually functions in the selenite assimilation pathway for selenoprotein biosynthesis in bacteria.

A reaction mechanism for alkylthiols with selenite has been proposed [35], in which alkylthioselenic acid (R-S-SeO_2_H) is generated first, then attacked by another alkylthiol, resulting in the formation of dithioselenite (R-S-Se(O)-S-R), which is further converted to the isomerized form (R-S-Se-O-S-R) [36]. Based on that mechanistic proposal [35,36] and the present results, we propose the early steps of the selenite delivery system with TrxA (Figure 5). First, selenite is attacked by the higher reactive thiol group of Cys33 to form thioselenite (Cys-S-SeO_2_H) which is suggested by the band shift of C36A caused by the incubation with selenite (Figure 2). Then, further nucleophilic attack by another thiol group of Cys36 results in the formation of dithioselenite (Cys-S-Se(O)-S-Cys), which is supported by our ESI-MS results (Figure 3). This TrxA–SeO complex can also be isomerized (Cys-S-Se-O-S-Cys), and these complexes could be dedicated to further reduction to selenide in later steps of the selenite delivery system. How the TrxA–SeO complex is further reduced to provide a selenide substrate for SPS remains an open question. Other TrxA molecules may be involved in the reduction of the TrxA–SeO complex, and the resulting oxidized TrxA would be reduced by TrxR in an NADPH-dependent manner, or alternatively, it may also be possible that TrxR directly reduces TrxA–SeO to produce selenide. Future studies will focus on a comprehensive understanding of the selenium delivery mechanisms.

## 4. Materials and Methods

### 4.1. Preparation of the Recombinant Proteins

To prepare His-tagged recombinant proteins, we constructed plasmids carrying the PsTrxA and PsTrxR genes as follows. The coding region of each gene (PszF2a_05700 for PsTrxA and PszF2a_19560 for PsTrxR) [27] was amplified from the genomic DNA of *P. stutzeri* F2a by PCR using the primer sets PsTrxA-f/PsTrxA-r and PsTrxR-f/PsTrxR-r for PsTrxA and PsTrxR, respectively (Appendix A). After digestion with NdeI (New England Biolabs, Ipswich, MA, USA) and BamHI (New England Biolabs), the fragments were individually inserted into the same restriction sites of pCold I (Takara Bio Inc., Kusatsu, Japan) to generate pCold-PsTrxA and pCold-PsTrxR. For site-directed mutagenesis, PCR proceeded using pCold-PsTrxA and the primer sets PsTrx_C33A-f/PsTrx_C33A-r and PsTrx_C36A-f/PsTrx_C36A-r for the PsTrxA mutants, C33A and C36A, respectively (Appendix A). After digestion of the template plasmid with DpnI (New England Biolabs), the mutated constructs were introduced into *E. coli* DH5α, resulting in pCold-PsTrxA_C33A and pCold-PsTrxA_C36A. The plasmid for gene expression of the PsTrxA C33A/C36A mutant, pCold-PsTrxA_C33A/C36A, was constructed using pCold-PsTrxA_C36A and the primer set PsTrx_C33A/C36A-f/PsTrx_C33A/C36A-r using the same procedure described above (Appendix A).

Plasmids for the expression of PsTrxA and its mutants were introduced into *E. coli* DH5α, and pCold-TrxR was introduced into *E. coli* BL21(DE3). The cells were grown at 37 °C in Luria–Bertani (LB) medium containing 100 μg mL^−1^ ampicillin (Nacalai Tesque, Kyoto, Japan) [37] until their optical density at 660 nm reached 0.4. The cells were cooled on ice for 30 min, then gene expression was induced by 0.2 mM isopropyl 1-thio-β-d-galactopyranoside (Protein Ark, Sheffield, UK), and the cells were further incubated at 16 °C for 24 h. The cells were harvested by centrifugation (10,000× *g*, 5 min, 4 °C), washed with phosphate-buffered saline [37], and collected again by centrifugation (10,000× *g*, 5 min, 4 °C). The cells were resuspended in 20 mM potassium phosphate (pH 7.4) containing 500 mM NaCl, 20 mM imidazole, and 5 mM 2-mercaptoethanol, sonicated, then centrifuged (15,000× *g*, 20 min, 4 °C). The crude extract was applied to a Ni-NTA Super Flow column (Thermo Fisher Scientific, Waltham, MA, USA), and the recombinant proteins were eluted by a stepwise increase in the imidazole concentration up to 500 mM. The purified proteins were buffer-exchanged to 40 mM potassium phosphate (pH 7.0) containing 5 mM 2-mercaptoethanol by ultrafiltration using an Amicon Ultra (Merck, Darmstadt, Germany). Protein concentration was determined using Protein Assay CBB Solution (Nacalai Tesque) by a Bradford protein assay [38].

### 4.2. Reduction Activity of the Recombinant Proteins

The disulfide-reducing activity of PsTrxA was evaluated by insulin reduction assays [28]. The reaction mixture (400 μL) contained 100 mM potassium phosphate (pH 7.0), 2 mM ethylenediaminetetraacetate, 150 μM insulin (Merck), 0.5 mM DTT, and 1 μM PsTrxA or its mutant protein. After preincubation at 37 °C for 3 min followed by addition of DTT and further incubation for 3 min, the reaction was initiated by adding PsTrxA proteins. The increase in absorbance at 650 nm due to the fact of precipitation of insulin by reductive cleavage of the disulfide bond by DTT reduced PsTrxA was measured.

The selenite-reducing activity of the Trx system of *P. stutzeri* F2a was assayed as follows. The reaction mixture (700 μL) contained 50 mM potassium phosphate (pH 7.0), 100 μM selenite, 300 μM NADPH, 2 μM PsTrxR, and 5 μM PsTrxA or its mutant protein. After preincubation at 37 °C for 5 min, the reaction was initiated by adding selenite, and the decrease in absorbance at 340 nm due to the decrease in NADPH was measured. The reducing activity towards other oxyanions was tested with 2 μM PsTrxA and 100 μM selenate, sulfite, sulfate, thiosulfate, nitrite, or nitrate instead of 100 μM selenite.

### 4.3. Cysteine–Thiol Group Labeling with PEG-PCMal

The numbers of free thiol groups in PsTrxA and its mutants were determined by labeling them with PEG-PCMal (Dojindo, Kumamoto, Japan) followed by SDS-PAGE. The PsTrxA and its mutants were reduced by incubation with 5 mM DTT in 50 mM potassium phosphate (pH 7.0) at 25 °C for 15 min. The mixture was applied to a Micro Bio-Spin 6 size exclusion column (Bio-Rad, Hercules, CA, USA) to remove DTT. The reduced PsTrxAs were incubated in a mixture containing 50 mM potassium phosphate (pH 7.0), 50 μM selenite, and 10 μM PsTrxA or its mutants at 25 °C for 15 min, followed by incubation with 1 mM PEG-PCMal at 37 °C for 20 min. Labeled samples were mixed with 17% (v/v) of loading buffer containing 10% SDS, 50% glycerol, 0.2 M tris(hydroxymethyl)aminomethane (Tris)-HCl (pH 6.8), and 0.05% bromophenol blue and separated on an 18% polyacrylamide gel by SDS-PAGE analysis.

### 4.4. ESI-MS Analysis

PsTrxA was DTT-reduced and incubated with selenite in the same manner as described in Section 4.3. The buffer of the PsTrxA mixture was replaced with sterile water to remove excess selenite using the Micro Bio-Spin 6. The protein samples were then mixed with the same volume of 98% methanol containing 2% formic acid and analyzed using a triple-quadrupole Sciex API 3000™ mass spectrometer (Applied Biosystems, Foster City, CA, USA) equipped with an electrospray ionization source in positive mode.

### 4.5. FDH Activity in Whole E. coli Cells

The FDH activity of the whole *E. coli* cells was examined as an indicator of selenoprotein biosynthesis using the strains, BW25113, JW5856-KC, and JW3564-KC as the wild type, Δ*EctrxA*, and Δ*selA* strains, respectively, from the Keio collection [39]. For complementation analysis of the decrease in FDH activity by PsTrxA and its mutants, the Δ*EctrxA* strain was transformed using pColdI, pCold-PsTrxA, pCold-PsTrxA_C33A, pCold-PsTrxA_C36A, or pCold-PsTrxA_C33A/C36A. The activity of FDH was assayed using the benzyl viologen agar overlay method [40]. *E. coli* cells were anaerobically cultivated overnight on LB solid medium containing 0.5% glucose at 37 °C. The medium was then overlayed with 0.75% agar containing 1.0 mg mL^−1^ benzyl viologen, 3.4 mg mL^−1^ KH_2_PO_4_, and 17 mg mL^−1^ sodium formate. The agar solidified within a few minutes, and cells with FDH activity were stained purple.

## Figures and Tables

**Figure 1 ijms-22-10965-f001:**
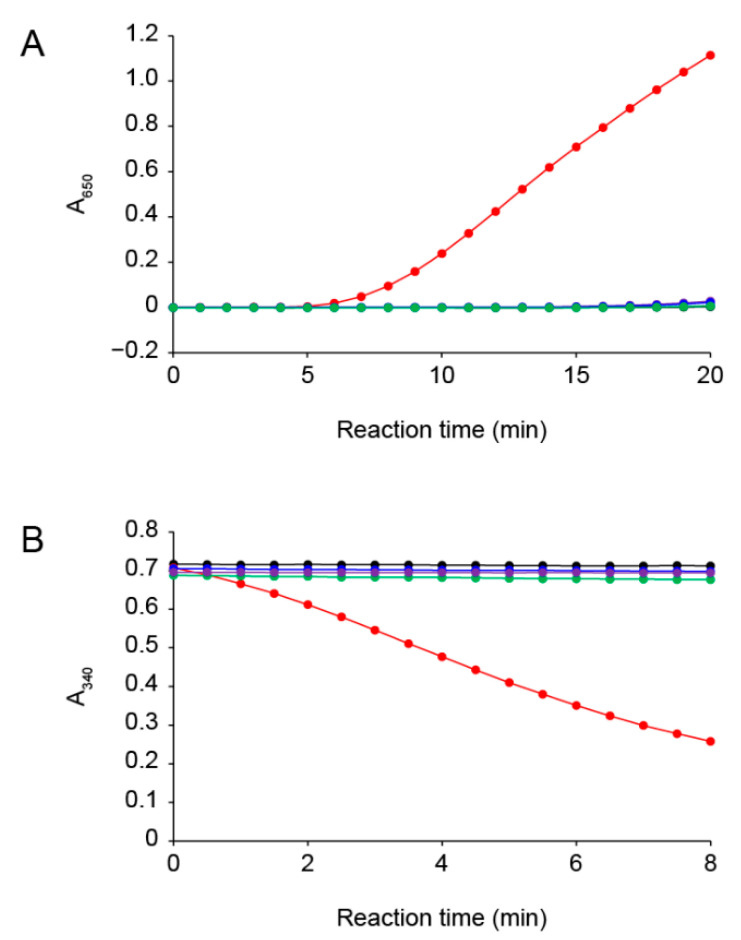
Reduction activity of the Trx system from *P. stutzeri* F2a. (**A**) Insulin disulfide-reducing activity of PsTrxA with DTT as measured by the increase in A_650_ due to the fact of insulin precipitation. (**B**) Selenite reducing activity of PsTrxA with PsTrxR and NADPH as measured by the decrease in A_340_ due to NADPH oxidation. PsTrxA proteins used in the assays were wild type (red), C33A (blue), C36A (green), and C33A/C36A (purple). Assays without PsTrxA are shown in black. Representative data obtained for each experiment are shown.

**Figure 2 ijms-22-10965-f002:**
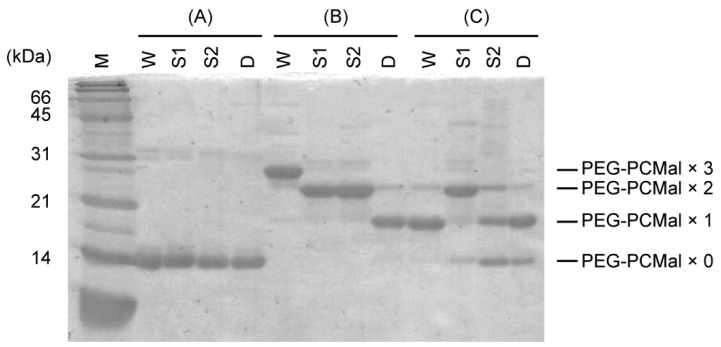
Band shifts of PsTrxAs on SDS-PAGE. PsTrxAs were labeled with PEG-PCMal depending on the numbers of free thiol groups. Recombinant PsTrxA proteins (**A**) were reduced by incubation with DTT, then DTT was removed by size exclusion chromatography. The resulting reduced proteins were incubated without (**B**) or with (**C**) selenite, labeled with PEG-PCMal, then resolved by SDS-PAGE. W, wild type; S1, C33A; S2, C36A; D, C33A/C36A.

**Figure 3 ijms-22-10965-f003:**
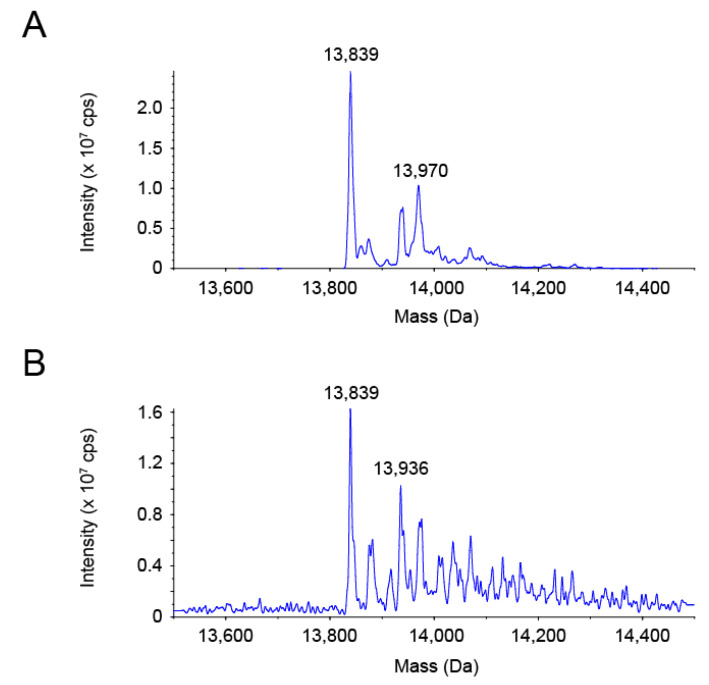
Reconstructed ESI-MS spectra of PsTrxA. PsTrxA protein was incubated without (**A**) or with (**B**) DTT and selenite, followed by ESI-MS analysis.

**Figure 4 ijms-22-10965-f004:**
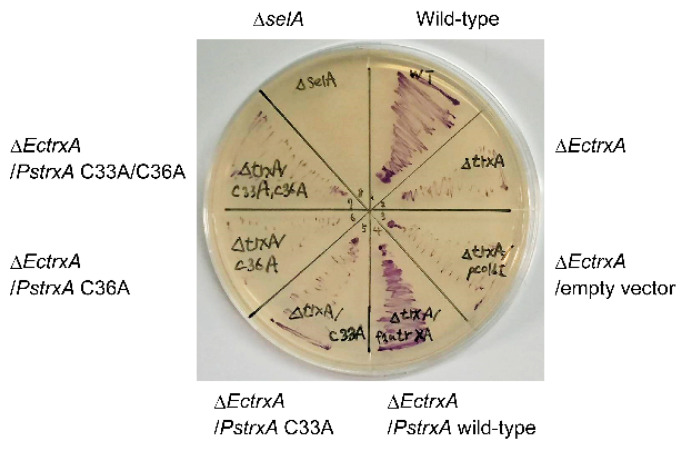
Whole cell FDH assay using *E. coli*. *E. coli* cells of the wild type, a SelA gene disruptant (Δ*selA*), and EcTrxA disruptants (Δ*EctrxA*) complemented without (none or empty vector) or with PsTrxA variants (wild type, C33A, C36A, or C33A/C36A) were anaerobically cultured on Luria–Bertani medium containing 0.5% glucose, then FDH activity was assayed using benzyl viologen.

**Figure 5 ijms-22-10965-f005:**
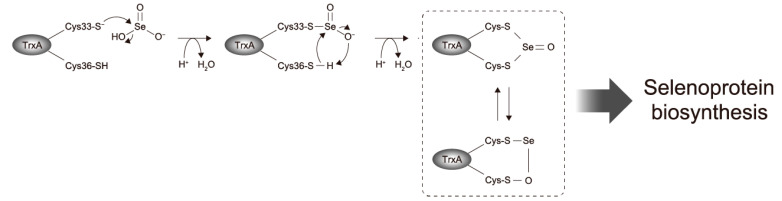
Proposed mechanism for early steps of selenite reduction via TrxA in the Trx system-dependent selenium delivery for selenoprotein biosynthesis.

## Data Availability

The data presented in this study are available upon request from the corresponding author.

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
