# Peer review of "Initial Step of Selenite Reduction via Thioredoxin for Bacterial Selenoprotein Biosynthesis"

_ijms, 2021, doi:10.3390/ijms222010965_

Round 1
Reviewer 1 Report
The purpose of the experiments presented in this manuscript was to evaluate the importance of the thioredoxine (Trx) and thioredoxine reductase (TrxR) system for the reduction of selenite into selenide and its incorporation into selenoproteins in the bacteria Pseudomonas stutzeri. In agreement with previous analyses conducted in E. coli, it was demonstrated the importance of TrxR and active TrxA for selenite reduction in this bacteria, and that this reduction was specific to selenite among other oxyanions studied. Experiment using PEG-PCMal labbeling indicated that the first nucleophilic attack takes place on Cys33 of TrxA, and was predicted to produce a covalent thioselenite intermediate PsTrxA-SeO. This intermediate was further characterized and validated by ESI-mass spectrometry. Finaly, it was shown that Pseudomonas stutzeri TrxA is able to complement the E coli homologous gene and to provide selenide for synthesis of the selenoprotein FDH. A mechanistic model for the selenite reduction by TrxA was proposed.
Overall, the experiments are well designed, and the methodological progression is logical. Data are accurately presented and discussed. However, the main question relates to the significance of information provided by these experiments compared to previous publications on the importance of the Trx system in selenite reduction in E coli (Takahata et al., 1985; Kumar et al., 1992 and Tamura et al., 2011). To improve the significance of the manuscript, it needs to be specified and explained why P. stutzeri enzymes rather than their E. coli homologs were selected for this study. Is it a rational for this?
It was previously shown that Trx and TrxR are required for selenite reduction in bacteria and a study using 75Se-labeled selenite demonstrated the formation of a selenite-TrxA adduct, but the mechanism of selenite reduction remained to be characterized. In this report, the authors characterized the Cys residues involved in the reaction and provide an initial characterization of the catalytic mechanism, based on the identification of one possible intermediate.
In conclusion, this work is potentially interesting and worthy of publication in IJMS.
Minor comment: in Figure 2, one should notice the presence of residual upper or lower bands in pannel (C), corresponding to minor Cys oxidation status, and explain how these bands can possibly be formed based on the proposed mechanism of reaction.
Author Response
We thank the reviewer for carefully examining our manuscript and for providing constructive comments. Our point-by-point responses to the reviewer’s comments are provided in attached file. Please see the attachment.

Reviewer 2 Report
The manuscript entitled “Initial step of selenite reduction via thioredoxin for bacterial selenoprotein biosynthesis” by Atsuki Shimizu and colleagues presents a set of convincing experiments and results on the initial biochemical conversion of selenite in bacteria, highlighting the central role of TrxA for this process. The methods have been described in sufficient detail, and in particular the complementary biological experiments with E coli transfected with PstrxA and the mutants nicely supports the conclusions drawn.
Major: -
Minor:
Introduction: It would be good to add a few words on the different roles of selenoproteins in bacteria and mammals, in order to better inform the reader on the central biochemical pathways that are controlled by selenoproteins in the different kingdoms of life. In addition, some mentioning on the different positions of SECIS elements in bacteria and mammals, and the analogies and differences would be good – some short and brief information would be sufficient.
Mammalian TrxR would currently be abbreviated with the gene and protein name TXNRD; please correct your nomenclature.
Results: The sentence “However, none of them was inert as a substrate (data not shown)” sounds strange. Is “inert” the right word here, or should it rather read: none of them was accepted as substrate? Why not presenting the results as Fig. 1C, in order to better highlighting the specificity of the system for selenite?
Figure 1: number of replicates studied is missing in the figure legend.
Discussion: As selenate is often used as inorganic Se source in culture, a few words on how selenate becomes converted to selenite would be welcome.
Materials and methods: 4.2. How was the concentration of PsTrxA (1 microM) be determined? Please explain.
References: Ref. 2 is pretty outdated – it may be worth to replace it with a more recent review on Se in human health.
Author Response

(The authors gave the same response as above.)
